# Critical Evaluation of Different Passive Sampler Materials and Approaches for the Recovery of SARS-CoV-2, Faecal-Indicator Viruses and Bacteria from Wastewater

Davey L. Jones [1,2,*], Jasmine M. S. Grimsley [3,4], Jessica L. Kevill [1], Rachel Williams [1], Cameron Pellett [1], Kathryn Lambert-Slosarska [1], Andrew C. Singer [5], Gwion B. Williams [1], Rafael Bargiela [1], Robert W. Brown [6], Matthew J. Wade [3] and Kata Farkas [1,7]

1　Centre for Environmental Biotechnology, Bangor University, Bangor LL57 2UW, UK
2　Food Futures Institute, Murdoch University, Murdoch 6150, Australia
3　Data, Analytics, and Surveillance Group, UK Health Security Agency, London SW1P 3JR, UK
4　The London Data Company, London EC2N 2AT, UK
5　UK Centre for Ecology & Hydrology, Wallingford OX10 8BB, UK
6　School of Natural Sciences, Bangor University, Bangor LL57 2UW, UK
7　School of Ocean Sciences, Bangor University, Menai Bridge, Anglesey LL59 5AB, UK
*　Correspondence: d.jones@bangor.ac.uk; Tel.: +44-7827–807516

**Abstract:** During the COVID-19 pandemic, wastewater-based epidemiology (WBE) has proven to be an effective tool for monitoring the prevalence of SARS-CoV-2 in urban communities. However, low-cost, simple, and reliable wastewater sampling techniques are still needed to promote the widespread adoption of WBE in many countries. Since their first use for public health surveillance in the 1950s, many types of passive samplers have been proposed, however, there have been few systematic studies comparing their ability to co-capture enveloped viruses and bacteria. Here, we evaluated the laboratory and field performance of 8 passive sampler materials (NanoCeram, ZetaPlus, nylon and ion exchange membranes, cellulose acetate filters, glass wool, cotton-based Moore swabs and tampons) to capture viruses and bacteria from wastewater. Viral capture focused on SARS-CoV-2, the bacteriophage Phi6 and the faecal marker virus, crAssphage. We showed that the best performing passive sampler in terms of cost, ease of deployment and viral capture were the electronegative cotton-based swabs and tampons. We speculate that viral capture is a combination of trapping of particulate matter to which viruses are attached, as well as electrostatic attraction of viral particles from solution. When deployed at wastewater treatment plants, the passive samplers worked best up to 6 h, after which they became saturated or exhibited a loss of virus, probably due to night-time wash-out. The patterns of viral capture across the different sampling materials were similar providing evidence that they can be used to monitor multiple public health targets. The types of bacteria trapped by the passive samplers were material-specific, but possessed a different 16S rRNA gene profile to the wastewater, suggesting preferential retention of specific bacteria. We conclude that the choice of passive sampler and deployment time greatly influences the pattern and amount of viral and bacterial capture.

**Keywords:** environmental monitoring; in situ sampling; public health surveillance; sewage network; viral capture method

## 1. Introduction

Wastewater based epidemiology (WBE) has the potential to simultaneously address a wide range of public health and environmental issues and is gaining traction worldwide as a mechanism to support public health decision making [1–3]. WBE has recently come to public attraction due to its widespread use for the routine monitoring of SARS-CoV-2 concentrations in wastewater and the use of this information to implement control measures

to help contain the spread of COVID-19 [4–6]. The effective use of WBE for informing policy and practice, however, relies very heavily on robust quality-assured methods for the collection and analysis of samples and subsequent reporting of data [7–9]. In the case of SARS-CoV-2, this has led to the progressive refining of existing technologies for recovering viral RNA from wastewater [10,11], as well as the implementation of a wide range of approaches to quantify and sequence the amount of SARS-CoV-2 RNA present in sewage [12–14].

To date, most studies on the use of WBE for community-level detection and control of SARS-CoV-2 have focused on sample collection at centralized wastewater treatment plants [5,15,16]. For that purpose, it is preferable to collect samples with a refrigerated autosampler rather than a one-off grab sample so that any diurnal variation in viral abundance in wastewater is captured [17]. However, if samples need to be taken within the sewer network to improve the spatial resolution of the surveillance strategy, this often requires taking samples from pumping stations or manholes, the latter often located in difficult to access public locations [18,19]. Under these circumstances, it is often impractical or too costly to deploy autosamplers, which may also be prone to vandalism and/or theft. This has led to the development of a range of simple, cost-effective, passive samplers that can be placed in the sewer water flow for varying lengths of time and then recovered for subsequent viral analysis [20].

The premise of most passive sampling devices is that they possess a charged surface which attracts and holds viruses, microbes and other charged solutes over a prolonged period of time [21]. In some cases passive sampling has proven better or comparable to conventional autosampler approaches [22,23]. In addition, passive samplers have no moving parts to fail, require no electricity to function, can be placed in any size of pipe, are suitable for confined spaces and can be located out of sight to avoid vandalism/theft.

Passive sampler technology has a long history within WBE and environmental surveillance [24–30], however, there have been relatively few technological innovations since their first introduction. These innovations include; the use of sampling membranes with contrasting physical and chemical properties to select for different analytes and the development of samples housings that facilitate easy deployment and which minimize pipe clogging/ragging [31,32].

If passive samplers are to be adopted for regional or national-scale surveillance, and subsequent decision making, it is vital that we better understand their performance under a wide range of conditions, enabling the development of standard operating procedures, ensuring repeatable performance, and allowing valid inter-site comparisons to be made. Therefore, the overall aim of this study was to critically evaluate the use of eight passive sampler materials, purported to be suitable for viral sorption, including the capture of SARS-CoV-2 and the human marker virus, crAssphage, from wastewater. Specifically, we aimed to (i) characterise the fundamental physicochemical properties of each different sorbent material, (ii) assess their performance under controlled laboratory conditions, (iii) deploy the best performing sampler materials within the sewer network and at centralized wastewater treatment plants for different time periods, (iv) assess the performance of different sampler housings, and (v) compare the passive sampler results to conventional autosampler approaches for viral quantification. While the eight passive sampler materials have been used in previous studies (Table 1), they have never been systematically compared against each other and not for simultaneous viral and bacterial capture, hence the novelty of this study.

**Table 1.** Summary of the eight different passive sampler materials used in the study.

| Sample | Material | Main Composition | Reference |
| --- | --- | --- | --- |
| A | NanoCeram®, Argonide Corporation | Patent protected | [33,34] |
| B | SG81 ion exchange paper, Whatman | Silica-cellulose | [35] |
| C | GW-40 glass wool | Glass | [36] |

**Table 1.** *Cont.*

| Sample | Material | Main Composition | Reference |
|---|---|---|---|
| D | ZetaPlus®, 3M | Patent protected | [37] |
| E | Cotton (Moore) swab | Cotton fibre | [28] |
| F | Biodyne® nylon membrane | Nylon | [38] |
| G | 11106, Sartorius membrane | Cellulose acetate | [39] |
| H | Inner tampon (Tampax Compk Super) | Cotton-based | [40] |
| I | Outer tampon (Tampax Compk Super) | Cotton-based | [40] |

## 2. Materials and Methods

### 2.1. Passive Sampler Selection

We chose eight contrasting passive sampler materials with surface charge for capturing viruses and viral nucleic acids. These materials were selected based on their previous use for capturing and concentrating either bacteria or viruses from wastewater, freshwater or marine waters (Table 1). The passive samplers were constructed from a range of materials, including mineral-based (e.g., NanoCeram, ZetaPlus), organic-based (e.g., cotton swab), mineral-organic composite-based (e.g., SG81) substances (Table 1). Photographs of the surface structure of the different sorbent materials used in the trial are provided in Figure S1.

### 2.2. Characterisation of Passive Sampler Materials

Total carbon and nitrogen content of each material were measured by mechanical grinding using a Retsch MM200 stainless steel ball mill (Retsch GmbH, Haan, Germany), followed by analysis using a Leco TruSpec C/N analyser (Leco Corp, St Joseph, MI). The point of zero charge ($pH_{pzc}$) was undertaken using the differential pH method [41]. The ash content of each material was determined after oven-drying the material at 80 °C (16 h) and then measuring the mass loss after placement in a programmable furnace at 450 °C for 16 h (Carbolite Gero Ltd, Hope Valley, UK). The pH and electrical conductivity (EC) of each material was determined in a 1:20 (*w*/*v*) deionized water extract using standard electrodes.

To determine the physical stability of each material, 0.5 g was placed in a 50 cm$^3$ polypropylene tube and shaken with 25 mL of deionized water (200 rev min$^{-1}$; 60 min). The amount of physical degradation of the material was determined by measuring the turbidity of the solution (300 μL) at 600 nm using a Synergy 96-well microplate reader (BioTek Inc., Winooski, VT).

Cation exchange capacity (CEC) was determined on 1 cm$^2$ pieces of material using an ammonium acetate procedure [42]. Briefly, samples were soaked in 1 M ammonium acetate (pH 7.0) overnight, washed several times with isopropyl alcohol and then leached with 1 M KCl. The amount of $NH_4^+$ leached from the material was determined colorimetrically using the salicylate reaction [43]. CEC was determined as the unit amount of $NH_4^+$ charge held per unit area (μmol cm$^{-2}$).

The anion retention exchange capacity of each sample was determined by placing 1 cm$^2$ in a 20 cm$^3$ polypropylene tube containing 10 mL of $^{33}P$-labelled $K_2HPO_4$ (10 mg P L$^{-1}$; pH 7.2; 0.49 kBq mL$^{-1}$). The tubes were then shaken (100 rev min$^{-1}$, 1 h) and the $^{33}P$ remaining in solution determined by removing 1 mL of the bathing solution and mixing with Optiphase HiSafe 3 scintillation fluid (Perkin-Elmer, Waltham, MA, USA) and measurement of $^{33}P$ by liquid scintillation counting using a Wallac 1404 liquid scintillation counter with automated quench correction (Wallac EG&G, Milton Keynes, UK). Phosphate sorption was determined as the unit amount of P held per unit area [44].

Protein-binding capacity was determined by placing 1 cm$^2$ of each material in a 20 mL polypropylene vial containing 10 mL of a mixture of uniformly $^{14}C$-labelled soluble proteins (0.2 kBq mL$^{-1}$; [45]). The samples were then shaken (100 rev min$^{-1}$; 1 h) and the $^{14}C$-labelled protein remaining in solution determined by removing 1 mL of the bathing solution and measuring its $^{14}C$ content by liquid scintillation counting as described above.



Water holding capacity was determined by placing 1 cm$^2$ of each material into 20 mL of distilled water and incubating overnight. The samples were then removed from the water and allowed to drain by gravity before measuring their water content by oven drying (105 °C, 16 h). Specific surface area of each material was performed on a Gemini Surface Area Analyser (Micromeritics Instrument Corporation, GA, USA) with nitrogen as the adsorbate, and liquid nitrogen as the sample coolant. The amount of gas adsorbed at a given pressure allowed for the calculation of the surface area utilizing the Brunauer-Emmett-Teller (BET) theory [46]. The chemical composition of the mineral based materials was undertaken using a S2 PicoFox TXRF spectrometer (Bruker Inc., Billerica, MA, USA).

*2.3. Viral Test Materials*

The enveloped virus *Pseudomonas* phage Phi6 (double-stranded RNA genome; 13.5 kb; DSM 21518) and the *Pseudomonas sp.* host (DSM 21482) were sourced from the Leibniz Institute DSMZ-German Collection of Microorganisms and Cell Cultures and cultivated in-house [47].

A concentrated stock of SARS-CoV-2 (Wuhan strain) propagated in cell culture was obtained from Andrew Weightman at Cardiff University. Prior to use, the virus was heat inactivated to mimic the non-infectious nature of the virus in wastewater and its different physical states [48] and to allow experimentation in a Biosafety Level 2 (BSL2) environment.

*2.4. Testing the Different Passive Sampler Materials under Laboratory Conditions*

Each passive sampler material possessed a slightly different thickness. Consequently, we normalized for this by using the same sorbent surface area in the experimental trials. Replicate samples of each passive sampler material (2 cm$^2$) were placed in pre-sterilized 50 cm$^3$ polypropylene tubes. Subsequently, wastewater containing heat-treated SARS-CoV-2 (Wuhan lineage; $10^7$ gc L$^{-1}$) or Phi6 ($10^9$ gc L$^{-1}$) were added. The samples were then gently shaken (30 rev min$^{-1}$) to maintain water movement. After 1 h, samples were removed for viral analysis by qPCR as described in Section 3.6 and 3.7. All of the controlled laboratory-based experiments used wastewater from the Bangor wastewater treatment plant (North Wales, operated by Dŵr Cymru-Welsh Water; Figure S2; [5]).

*2.5. Testing the Different Passive Sampler Materials under Field Conditions*

For the field deployment trials, we used the centralized urban wastewater treatment plant located in Bangor and operated by Dŵr Cymru-Welsh Water (Figure S2). The mean flow at the Bangor wastewater treatment plants is 108 l s$^{-1}$, respectively. At this site, the passive samplers were placed behind the crude influent screen, unless otherwise stated, with trials undertaken between May and August 2021 (Figure S3). All sample processing was conducted in a BSL2 laboratory, adhering to WHO and national biosafety guidelines. Three field deployment trials were conducted (i) to select the best performing materials, (ii) to determine optimal deployment times, and (iii) to assess the usefulness of protective housing for the passive samplers.

2.5.1. Field Trial 1-Material Selection

In the first field trial, replicate squares of each passive sampler material (5 × 5 cm$^2$) were placed in polypropylene mesh cages (Figure S4) and placed after the crude inflow grate at the Bangor wastewater treatment plant at 11.00 am (start of peak flow). The samples were recovered after either 1 or 24 h, placed in a refrigerated cool box and transported back to the laboratory for sample processing. For comparison, a composite water sample was also taken using a dip sampler over the first hour (6 × 150 mL taken every 10 min). On arrival, the passive sampler membranes were cut in half. The first half was used to quantify the concentration SARS-CoV-2 and the human faecal marker, crAssphage, as detailed in Sections 2.6 and 2.7. The second half was used to assess the bacterial communities on the passive samplers using 16S rRNA sequencing (Section 2.8).

### 2.5.2. Field Trial 2-Deployment Time Selection

In the second field trial, a subset of the best performing passive sampler materials (B, C, E and H; Table 1) were deployed in triplicate for different time periods (0.5, 1, 3, 6 and 24 h) to determine the optimal time for deployment (Figure S5). These were placed in a sewer line feeding into the wastewater treatment plant at Bangor which leads from the local hospital (53°12′31.71″ N, 4°9′35.76″ W) and surrounding urban area (ca. 1 km away). The sewer was not screened allowing us to evaluate whether the passive samplers would accumulate debris (Figure S6). The passive samplers were removed at known times and refrigerated prior to analysis for SARS-CoV-2 and crAssphage. In addition, water samples were taken at 10 min intervals to assess changes in wastewater chemistry (e.g., pH, EC, $NH_4^+$, P, turbidity) during the deployment. For logistical reasons, only two types of passive sampler materials were evaluated at any one time. On return to the laboratory, half the sampler membrane was used for viral analysis and the remainder used to assess particle capture. For particle capture, the passive sampler membrane was placed in 25 mL of distilled water and mechanically shaken at 250 rev min$^{-1}$ for 1 h. The amount of particulates displaced from the passive sampler into the bathing solution was then measured by removing the membrane and oven-drying the solution (105 °C, 16 h) to determine the dry weight of solids remaining.

### 2.5.3. Field Trial 3-Housing Selection

In the final set of trials, we evaluated the performance of the Monash torpedo sampler for housing the sampler membrane [32] versus a membrane cassette floating freely in the wastewater stream (Figure S6). The samplers were deployed in the unscreened hospital-fed sewer line entering the Bangor wastewater treatment plant as described above (see Section 2.5.2) for either 1 h or 24 h after which they were recovered for visual inspection, viral quantification and particle accumulation.

### *2.6. Viral RNA/DNA Recovery and Extraction*

A 1 cm$^2$ area of each passive sampler was cut off using sterile scissors and forceps and placed in 15 mL centrifuge tubes containing 2 mL of NucliSens lysis buffer (BioMerieux, Marcy-l'Étoile, France; Cat No. 280134 or 200292). The tubes were then vortexed and prepared for manual nucleic acid extraction as detailed previously [49].

Viruses in the water samples were concentrated using a polyethylene glycol-based precipitation method as detailed previously [49]. The water concentrates were eluted in 0.8 mL of NucliSens lysis buffer and extracted using Kingfisher 96 Flex system (Thermo Scientific, Waltham, MA, USA), as detailed previously [47]. The final volume of the nucleic acid extracts was 0.1 mL.

### *2.7. RT-qPCR Analysis*

The (RT-)qPCR assays were carried out using the QuantStudio® Flex 6 Real-Time PCR System (Applied Biosystems, Waltham, MA, USA). For RNA targets, one-step RT-qPCR assays were used with well-established primer and probe sets for the N1 gene region of SARS-CoV-2 [50] and the 6S_1 gene region of the Phi6 bacteriophage genome [51] using the 1x TaqMan™ Fast Virus 1-Step Master Mix (Applied Biosystems, Waltham, MA, USA) as described elsewhere [52]. Synthetic RNA of the target region was used for the preparation of standard dilution series for quantification (Table S1). Molecular grade water was used as non-template control, which was negative in all experiments.

The qPCR for crAssphage was performed using a QuantiFast probe PCR assay (Qiagen, Hilden, Germany) as described previously [47]. Plasmid DNA as described elsewhere [53] was used for quantification (Table S1). Molecular grade water was used as non-template control, which was negative in all experiments.

The (RT-)qPCR data was converted to gc μL$^{-1}$ nucleic acid extract, gc L$^{-1}$ wastewater and gc cm$^{-2}$ for passive samplers for statistical analysis. The limit of detection (LOD) was tested using 10 replicate dilutions of N1 and crAssphage genomic RNA/DNA, and defined

as the minimum concentration whereby 10 replicates all return positive results: 1.7 gc $\mu L^{-1}$ for N1 and 2 gc $\mu L^{-1}$ crAssphage. As such, quantities can be detected below this limit but are susceptible to false negatives. The viral load of seeded SARS-CoV-2 was confirmed by RT-qPCR and this number was used to calculate the percent recovery for samples in the controlled experiment.

*2.8. DNA Extraction, 16S rRNA Gene Amplicon Sequencing and Analysis*

Bacterial DNA was extracted from each passive sampler using the Zymo BIOMICS DNA Miniprep Kit (Zymo Research, Irvine, CA, USA) according to manufacturer's instructions. Two independent DNA extractions were carried out with a high-speed bead beating for each sample. Quality and concentration of extracted DNA were assessed by gel electrophoresis and by Qubit 4.0 Fluorometer dsDNA BR Assay Kit (Life Technologies, Carlsbad, CA, USA). Libraries of 16S rRNA gene amplicons were prepared by single PCR with double-indexed fusion primers as described previously [54]. Hypervariable V4 16S rRNA gene fragment was amplified using modified forward primer F515 (5′-GTGBCAGCMGCCGCGGTAA-3′) and reverse R806 prokaryotic primer (5′-GGACTAC HVGGGTWTCTAAT-3′), which amplify an approximately 290 bp region. Primers were designed to contain: the Illumina adapters and sequencing primers, a 12 bp barcode sequence, a heterogeneity spacer to mitigate the low sequence diversity amplicon issue, and 16S rRNA gene universal primers [54]. PCRs were performed using OneTaq DNA Polymerase (New England Biolabs, Ipswich, MA, USA). All reactions were run with no-template negative controls. Thermocycling conditions were: initial denaturation at 95 °C for 2 min, followed by 30 cycles at 95 °C for 30 s, 50 °C for 50 s, and 72 °C for 90 s with a final elongation at 72 °C for 5 min. Amplicons were visualized in a 1.5% tris-acetate agarose gels using a GelDoc System (Bio-Rad, Hercules, CA, USA). DNA bands of approximately 440 bp were gel-purified using QIAEX II Gel Extraction Kit (Qiagen). The purified amplicons were then quantified using Qubit 4.0 Fluorometer, pooled in equimolar amounts and the final pool was run on Illumina MiSeq platform (Illumina, San Diego, CA, USA) using 500-cycle v2 chemistry ($2 \times 250$ bp paired-end reads) at the Centre for Environmental Biotechnology, Bangor, UK.

Raw sequencing reads were processed according to previously described protocols [54,55]. Briefly, the data was pre-processed to extract the barcodes from sequences, and then cleaned of primer sequences using tagcleaner. The barcodes and the sequences were re-matched again using in-house Python scripts. The resulting filtered reads were analysed using QIIME v1.3.1. First, the libraries were demultiplexed based on the different barcodes. Then, the sequences were classified on operational taxonomic units (OTUs) combining de novo and reference-based methods (open-reference OTU generation algorithm) using the SILVA version 132 reference database.

*2.9. Statistical Analysis*

The data was checked for normality using a Kolmogorov–Smirnov test in Minitab v19 (Minitab Inc., State College, PA, USA). When it did not conform, the data was $\log_{10}$ transformed prior to analysis. Comparisons between treatments were undertaken by performing one-way ANOVAs with Tukey pair-wise comparisons in Minitab v19. Linear regression analysis was undertaken in SigmaPlot v14.5 (Systat Software Inc., Bath, UK). $p < 0.05$ was used as the cut-off value for statistical significance unless otherwise stated.

## 3. Results

*3.1. Characterisation of the Passive Sampler Materials*

A summary of the main chemical and physical properties of the passive sampler materials prior to wastewater deployment is shown in Tables 2–4. As expected, the materials had a contrasting range of properties. The intrinsic pH of the materials ranged widely from pH 4.1 to pH 10.7. All the samplers contained appreciable amounts of leachable salts, as evidenced by the EC values (80 to 420 $\mu S\ cm^{-1}$). The glass wool was the sole

mineral-based material which consisted of a K/Ca-silicate glass (Table 2). The NanoCeram, ZetaPlus and SG81 filter paper materials were organo-mineral composites as evidenced by their high organic C contents and large ash content, for which each material had a unique elemental composition. None of the passive samplers contained appreciable quantities of N suggesting that amino-groups probably do not contribute appreciably to the electrostatic properties of the materials.

**Table 2.** Major inorganic constituents of the mineral and organo-mineral based passive sampler materials. Values represent means $\pm$ SEM ($n = 3$) except for the glass wool which is $n = 1$.

| Sampler | Si | K | Ca | Fe | Zn |
|---|---|---|---|---|---|
| NanoCeram® | 47.6 $\pm$ 3.0 | 6.3 $\pm$ 0.1 | 3.7 $\pm$ 0.1 | 0.70 $\pm$ 0.06 | 15.7 $\pm$ 1.3 |
| SG81, Whatman | 99.8 $\pm$ 0.1 | <0.02 | 0.05 $\pm$ 0.01 | <0.01 | <0.01 |
| GW-40 glass wool | 76.6 | 0.7 | 20.1 | 2.0 | <0.01 |
| ZetaPlus®, 3M | 85.2 $\pm$ 0.1 | 7.1 $\pm$ 1.7 | 1.2 $\pm$ 0.2 | 2.0 $\pm$ 0.1 | <0.01 |

Of the materials tested, only the NanoCeram and Biodyne materials had an appreciable capacity to retain anions (i.e., orthophosphate; electropositive sites) with some materials (e.g., glass wool, cotton-based samplers) possessing very little capacity to retain P (Table 3). The ZetaPlus material had a very high cation binding capacity (i.e., $NH_4^+$ retention; electronegative), followed by NanoCeram, while most of the other materials possessed a low $NH_4^+$ retention capacity. In contrast to P and $NH_4^+$, all materials except glass wool, had the ability to retain soluble protein (Table 4). After placement in water and agitation, most of the materials retained their physical integrity except for the SG81 filter paper and, to a lesser extent, the NanoCeram filter. The point of zero charge (PZC) for all materials was below pH 7.0 with exception of the glass wool. It was not possible to accurately determine the surface area of some materials (below detection limit; < 0.01 m$^2$ g$^{-1}$), however, the SG81 and NanoCeram filters gave the highest values.

### 3.2. Viral Recovery by the Passive Sampler Materials under Laboratory Conditions

The recovery of the process control virus Phi6 from wastewater after one hour of incubation using the different passive sampler materials is presented in Figure 1A. Overall, there were significant differences in viral recovery between the different materials ($p < 0.001$). The best recovery of Phi6 was achieved with the tampon material and glass wool while no detectable Phi6 was retained by the NanoCeram material.

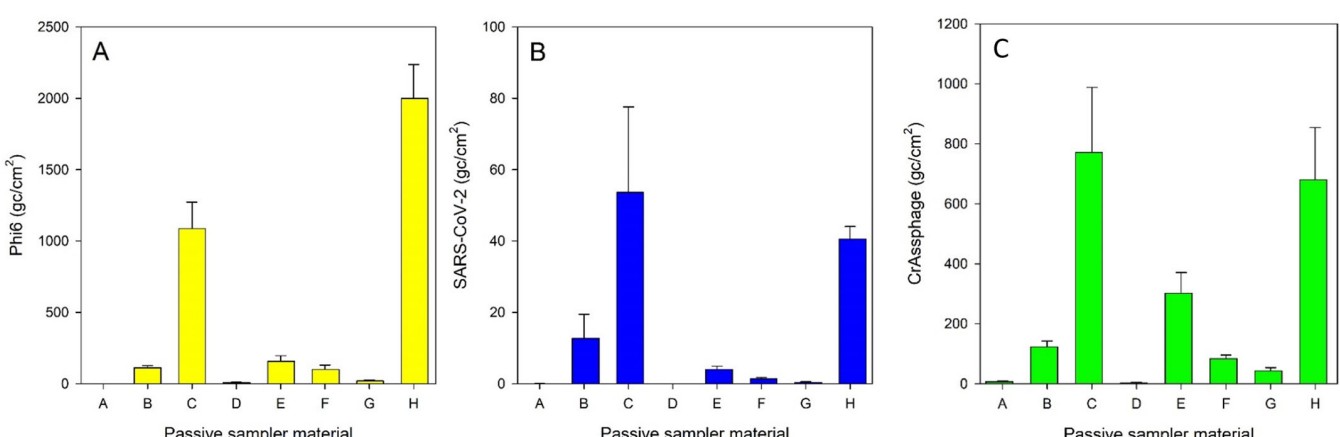

**Figure 1.** Recovery of the (**A**) enveloped virus Phi6, (**B**) SARS-CoV-2 RNA, and (**C**) the faecal marker crAssphage from spiked wastewater by 8 different passive sampler materials after incubation for 1 h. Values represent means $\pm$ standard error (SEM) ($n = 3$). The alphabetical key to the different passive sampler materials is presented in Table 1. gc: genome copies.

**Table 3.** Basic physical and chemical characteristics of the passive sampler materials used in the trials. Values represent means $\pm$ SEM ($n$ = 3).

| Sampler | pH | EC ($\mu$S cm$^{-1}$) | Moisture Content (%) | Organic Content (%) | Total C (%) | Total N (%) | P Sorption ($\mu$mol cm$^{-2}$) | NH$_4^+$ Sorption ($\mu$mol cm$^{-2}$) |
|---|---|---|---|---|---|---|---|---|
| NanoCeram® | 6.63 $\pm$ 0.23 | 424 $\pm$ 84 | 4.5 $\pm$ 0.1 | 50.7 $\pm$ 0.2 | 31.7 $\pm$ 0.1 | 0.12 $\pm$ 0.01 | 31.3 $\pm$ 1.2 | 12.9 $\pm$ 1.4 |
| SG81, Whatman | 6.50 $\pm$ 0.08 | 152 $\pm$ 11 | 5.3 $\pm$ 0.1 | 78.6 $\pm$ 0.1 | 34.1 $\pm$ 0.1 | 0.18 $\pm$ 0.06 | 0.8 $\pm$ 0.1 | 2.2 $\pm$ 0.7 |
| GW-40 glass wool | 10.7 $\pm$ 0.07 | 132 $\pm$ 8 | 0.4 $\pm$ 0.1 | 0.0 $\pm$ 0.0 | 0.0 $\pm$ 0.0 | 0.00 $\pm$ 0.00 | <0.1 | 0.2 $\pm$ 0.1 |
| ZetaPlus®, 3M | 7.08 $\pm$ 0.28 | 201 $\pm$ 34 | 3.0 $\pm$ 0.1 | 53.5 $\pm$ 0.3 | 23.9 $\pm$ 0.7 | 0.26 $\pm$ 0.00 | 0.5 $\pm$ 0.1 | 136.0 $\pm$ 35.9 |
| Cotton (Moore) swab | 6.33 $\pm$ 0.07 | 108 $\pm$ 8 | 4.9 $\pm$ 0.1 | 99.7 $\pm$ 0.1 | 51.3 $\pm$ 0.1 | 0.19 $\pm$ 0.01 | <0.1 | 0.7 $\pm$ 0.3 |
| Biodyne® nylon | 4.07 $\pm$ 0.24 | 88 $\pm$ 7 | 2.8 $\pm$ 0.1 | 99.7 $\pm$ 0.1 | 43.3 $\pm$ 0.1 | 0.01 $\pm$ 0.00 | 3.5 $\pm$ 0.1 | 0.3 $\pm$ 0.1 |
| 11106, Sartorius | 4.34 $\pm$ 0.01 | 311 $\pm$ 61 | 7.3 $\pm$ 1.8 | 99.7 $\pm$ 0.2 | 49.8 $\pm$ 0.4 | 0.05 $\pm$ 0.00 | 0.3 $\pm$ 0.0 | 0.2 $\pm$ 0.1 |
| Inner tampon | 6.65 $\pm$ 0.05 | 123 $\pm$ 16 | 3.7 $\pm$ 0.1 | 99.8 $\pm$ 0.0 | 53.7 $\pm$ 0.4 | 0.12 $\pm$ 0.03 | 0.3 $\pm$ 0.0 | 0.8 $\pm$ 0.1 |
| Outer tampon | 8.18 $\pm$ 0.05 | 181 $\pm$ 22 | 4.5 $\pm$ 0.2 | 99.7 $\pm$ 0.0 | 41.3 $\pm$ 0.3 | 0.18 $\pm$ 0.02 | <0.1 | 2.6 $\pm$ 0.1 |

**Table 4.** Basic physical and chemical characteristics of the passive sampler materials used in the trials. Values represent means $\pm$ SEM ($n$ = 3).

| Sampler | Protein binding (% of Total Added) | Physical Integrity (RAU) | Mass/Area Ratio (mg cm$^{-2}$) | PZC (pH Value) | Water Holding Capacity (g g$^{-1}$) | Specific Surface Area (m$^2$ g$^{-1}$) |
|---|---|---|---|---|---|---|
| NanoCeram® | 38.3 $\pm$ 2.2 | 382 $\pm$ 18 | 58 $\pm$ 1 | 6.26 $\pm$ 0.09 | 3.9 $\pm$ 0.2 | 19.0 $\pm$ 0.8 |
| SG81, Whatman | 22.0 $\pm$ 0.7 | 936 $\pm$ 330 | 21 $\pm$ 0 | 5.91 $\pm$ 0.08 | 5.5 $\pm$ 0.5 | 42.5 $\pm$ 1.4 |
| GW-40 glass wool | 0.4 $\pm$ 0.3 | 22 $\pm$ 11 | 71 $\pm$ 6 | 9.78 $\pm$ 0.01 | 6.5 $\pm$ 1.0 | <0.01 |
| ZetaPlus®, 3M | 53.3 $\pm$ 1.9 | 53 $\pm$ 17 | 248 $\pm$ 2 | 5.35 $\pm$ 0.03 | 6.5 $\pm$ 0.5 | 0.02 |
| Cotton swab | 24.1 $\pm$ 1.7 | 41 $\pm$ 19 | 28 $\pm$ 1 | 4.90 $\pm$ 0.04 | 21.2 $\pm$ 1.2 | <0.01 |
| Biodyne® nylon | 32.3 $\pm$ 0.5 | 4 $\pm$ 1 | 98 $\pm$ 6 | 6.02 $\pm$ 0.10 | 3.8 $\pm$ 0.5 | 3.8 $\pm$ 0.9 |
| 11106, Sartorius | 16.9 $\pm$ 0.8 | 2 $\pm$ 1 | 14 $\pm$ 1 | 4.54 $\pm$ 0.02 | 7.1 $\pm$ 0.3 | 7.0 $\pm$ 2.0 |
| Inner tampon | 40.3 $\pm$ 2.5 | 4 $\pm$ 1 | 153 $\pm$ 10 | 4.60 $\pm$ 0.04 | 24.2 $\pm$ 4.5 | <0.01 |
| Outer tampon | 22.2 $\pm$ 0.5 | 21 $\pm$ 5 | 33 $\pm$ 3 | 4.66 $\pm$ 0.06 | 12.9 $\pm$ 1.1 | <0.01 |

The results for the capture of SARS-CoV-2 from wastewater by the eight different passive sampler materials correlated well to those of Phi6 in terms of viral recovery ($r^2 = 0.74$) with significant differences apparent between the materials ($p < 0.001$; Figure 1B). The materials with the highest SARS-CoV-2 recovery were the tampon material, glass wool and the SG81 filter.

The recovery of the human faecal marker virus, crAssphage, which is naturally present in the wastewater, by the different passive sampler materials is presented in Figure 1C. Overall, the pattern of recovery was similar to that of SARS-CoV-2 and Phi6, except that a small amount of virus was recovered from the NanoCeram material while the cotton (Moore) swab also showed appreciable retention and recovery of the virus. Again, the best recovery of virus was seen with the glass wool and tampon material ($p < 0.001$).

### 3.3. Viral Recovery of the Different Passive Sampler Materials in Field Trials

Visual images of the surface of the eight different passive sampler materials after deployment at a centralized wastewater treatment plant for either 1 or 24 h is shown in Figure S7. Overall, the progressive surface accumulation of particles over time can be seen for all passive sampler materials. In particular, the glass wool trapped a lot of particulate matter in its matrix, while the smoother membrane surfaces (e.g., Biodyne filter) accumulated relatively few particles. The physical integrity of all materials was maintained throughout the 24 h deployment period. Unfortunately, none of the passive samplers consistently tested positive for the presence of SARS-CoV-2 N1 gene, presumably due to relatively low prevalence rates of COVID-19 in the local community during the time which the experiment was undertaken (52 cases per 100,000 people for this trial; Figure S3). The lack of viral RNA in the passive samplers was also reflected in the low prevalence of SARS-CoV-2 in the wastewater during the sampling period (<100 gc l$^{-1}$; data not presented).

In contrast to the static laboratory trials, all the passive samplers accumulated crAssphage within 1 h of deployment in flowing wastewater although differences were seen between the individual materials ($p < 0.001$; Figure 2). We ascribe the difference in accumulation pattern between the laboratory and field trials to the physical trapping/attachment of large amounts of particles containing crAssphage in the passive sampler matrix under flowing conditions as well as being absorbed and recovered from the surface.

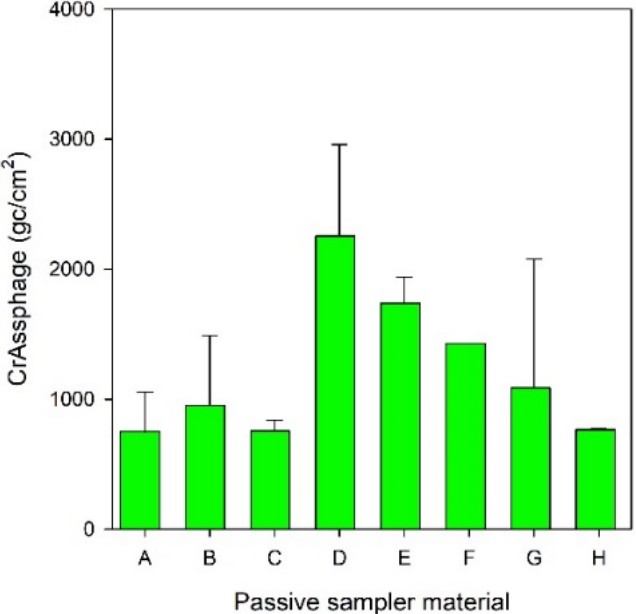

**Figure 2.** Recovery of the faecal marker virus crAssphage from wastewater by 8 different passive sampler materials after incubation in the field for 1 h. Values represent means ± SEM (*n* = 3). The alphabetical key to the different passive sampler materials is presented in Table 1. gc: genome copies.

### 3.4. Profiling Bacterial Community Composition from Different Passive Sampler Materials and Deployment Times

The 16S rRNA profile of the different passive samples collected after 1 or 24 h deployment at the Bangor wastewater treatment plant is presented in Figure 3. The results indicate that the bacterial profile captured by the NanoCeram passive filter (A) was very different from all the other materials, particularly after a 1 h deployment time. Generally, the 16S rRNA gene profile was dominated by Gammaproteobacteria and to a lesser extent Campylobacteria and Bacteroidia after 1 h, with many other bacterial classes also present. The profile of most passive samplers was similar to that of the liquid wastewater after 1 h. It is also clear that deployment time had a significant influence on the 16S rRNA gene profile when comparing the 1 h and 24 h profiles.

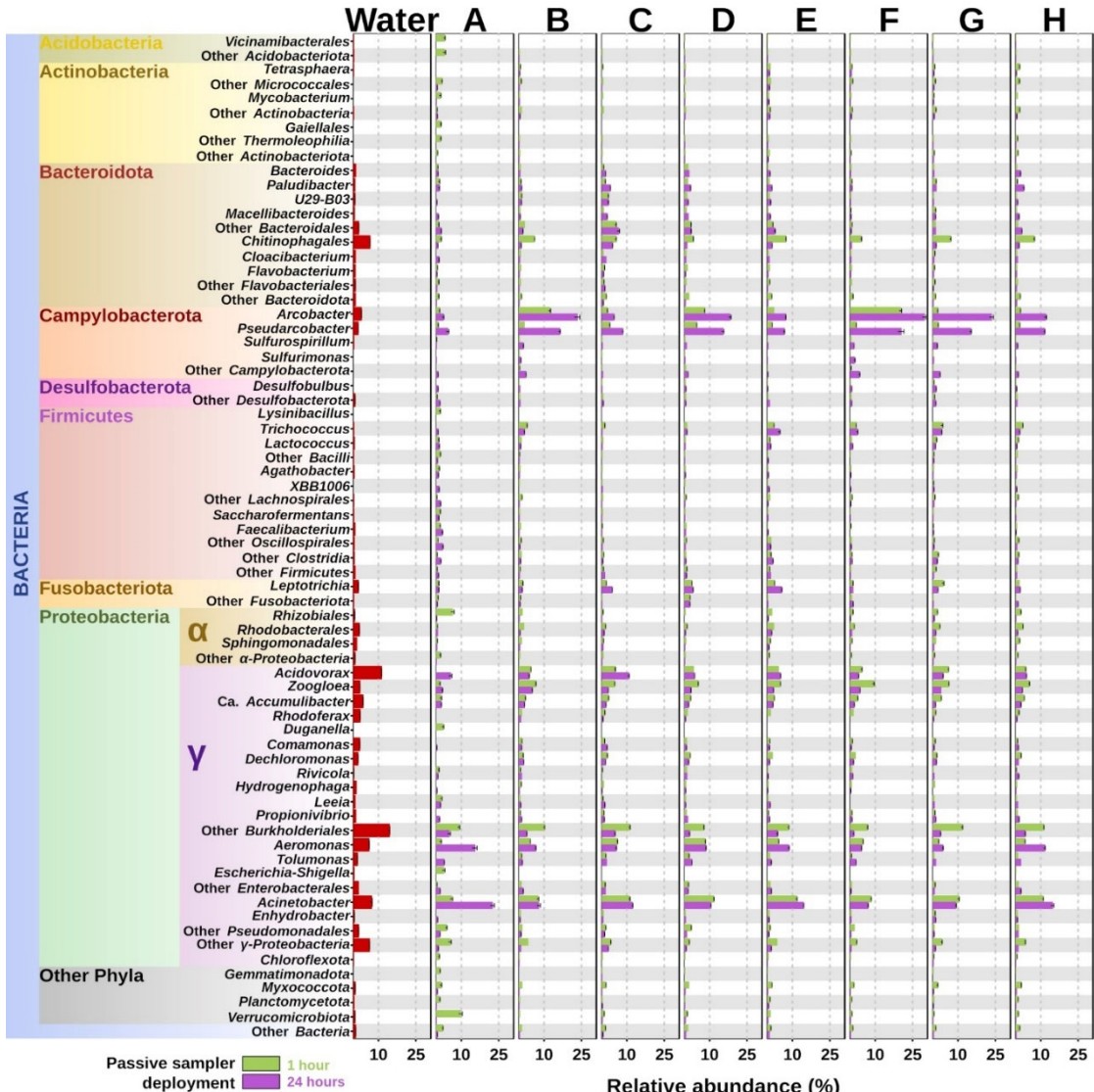

**Figure 3.** 16S rRNA bacterial diversity profile for eight different passive sampler materials (A to H) after deployment in the crude influent stream for either 1 or 24 h at a centralized wastewater treatment plant. The bacterial class profile for the three individual replicates for each sampler material are shown. The bacterial diversity of a composite sample of wastewater taken over the first hour of sampling is shown for reference at the left-hand side.

### 3.5. Influence of Deployment Time on Passive Sampler Efficiency

Based on the results of the viral capture trials described above, we selected the four best performing materials (i.e., glass wool, SG81 filter membrane, cotton swab, cotton

tampon) for further experimentation. The effect of deployment time (up to 24 h) on discoloration of the sampler material and the accumulation of particulate matter is shown in Figure S8. Visually, the glass wool and tampon materials continued to accrue particulate matter throughout the 24 h deployment period. In contrast, the SG81 filter appeared to accumulate much less material than the others, while the cotton (Moore) swab appeared to discolor quickly and visually accrue little particulate matter after 6 h. Measurement of the amount of particulate matter present on the passive samplers after their recovery from the wastewater stream supported the visual observations, showing a progressive accumulation in three of the samplers with the exception of the cotton (Moore) swab which peaked at 6 h (Figure S9). As the passive samplers were deployed on different days for logistical reasons, the amounts of particle accumulation between the two sets of data cannot be directly compared.

Measurements of wastewater chemistry over the first 6 h of deployment revealed some differences between the two deployment periods (Figure S10). However, the properties of the wastewater (pH, EC, $NH_4^+$, P, turbidity) remained relatively constant during the first 6 h of the passive sampler deployment period. This indicates that the accumulation in particulate matter was unlikely to be due to major changes in wastewater composition over time. The diurnal flow pattern for the periods of passive sampler deployment is shown in Figure S11. This clearly demonstrates the peak flow during the daytime period and a lower of flow at night. The mean flow on deployment period 1 was $70.1 \pm 4.3$ L s$^{-1}$, while on deployment period 2 the mean flow was $69.2 \pm 4.5$ L s$^{-1}$. The flow profile indicates that the first hour of passive sampling caught the morning flow peak.

All the passive sampler materials accumulated both P and $NH_4^+$ over time (Figure S12). In the case of the glass wool, this accumulation reflected the temporal pattern of particle trapping within the matrix. This pattern, however, was not reflected in the SG81 filter, which showed a much lower but progressive accumulation of both N and P. The cotton tampon showed a rapid accumulation of N and P which then increased slightly over the 24 h deployment period. In contrast, the cotton (Moore) swab accumulated N and P to a lesser extent than the tampon, and no progressive accumulation was seen during the deployment period. Overall, the results suggest that N and P capture by the passive sampler does not provide a good proxy of wastewater exposure time.

The influence of deployment time on the accumulation of SARS-CoV-2 and crAssphage by the four best passive sampler materials (cotton tampon, cotton swab, glass wool, SG81 filter) identified in the initial trials is shown in Figure 4. During the study period for this trial the regional prevalence of COVID-19 cases ranged from 99 to 109 cases per 100,000 in the catchment area. Overall, all four materials were able to capture SARS-CoV-2, however, the temporal dynamics were different for each material. The results indicate that the optimal deployment time for the passive samplers varies from 3–6 h, with no benefit in viral accumulation seen over longer times (e.g., 24 h). The dynamics of crAssphage capture were similar to those of SARS-CoV-2 except that the capture was ca. 2000-fold greater. The results for crAssphage also indicated that short deployment times are probably preferable for viral capture under high flow conditions typically seen at wastewater treatments plants (longer time periods may lead to saturation of the sampler).

### 3.6. Performance of the Different Sorbent Materials in the Torpedo Sampler

Overall, this preliminary trial suggested that the Monash torpedo was suitable for deployment in the network. Due to the low prevalence of SARS-CoV-2 in the community it was not possible to evaluate the relative ability to capture viruses. However, the amount of particles trapped by the passive samplers inside the Monash torpedo was less than those deployed directly into the wastewater when measured by two independent methods (Table S1). It should be noted that the Monash torpedo was only partially submersed in the top of the wastewater stream. Although this did not result in ragging issues, even after 24 h (Figure S6), it does imply that the exposure of the passive samplers to wastewater may

have been slightly altered both by spatial positioning in the water column and physical resistance of water entering the holes in the device.

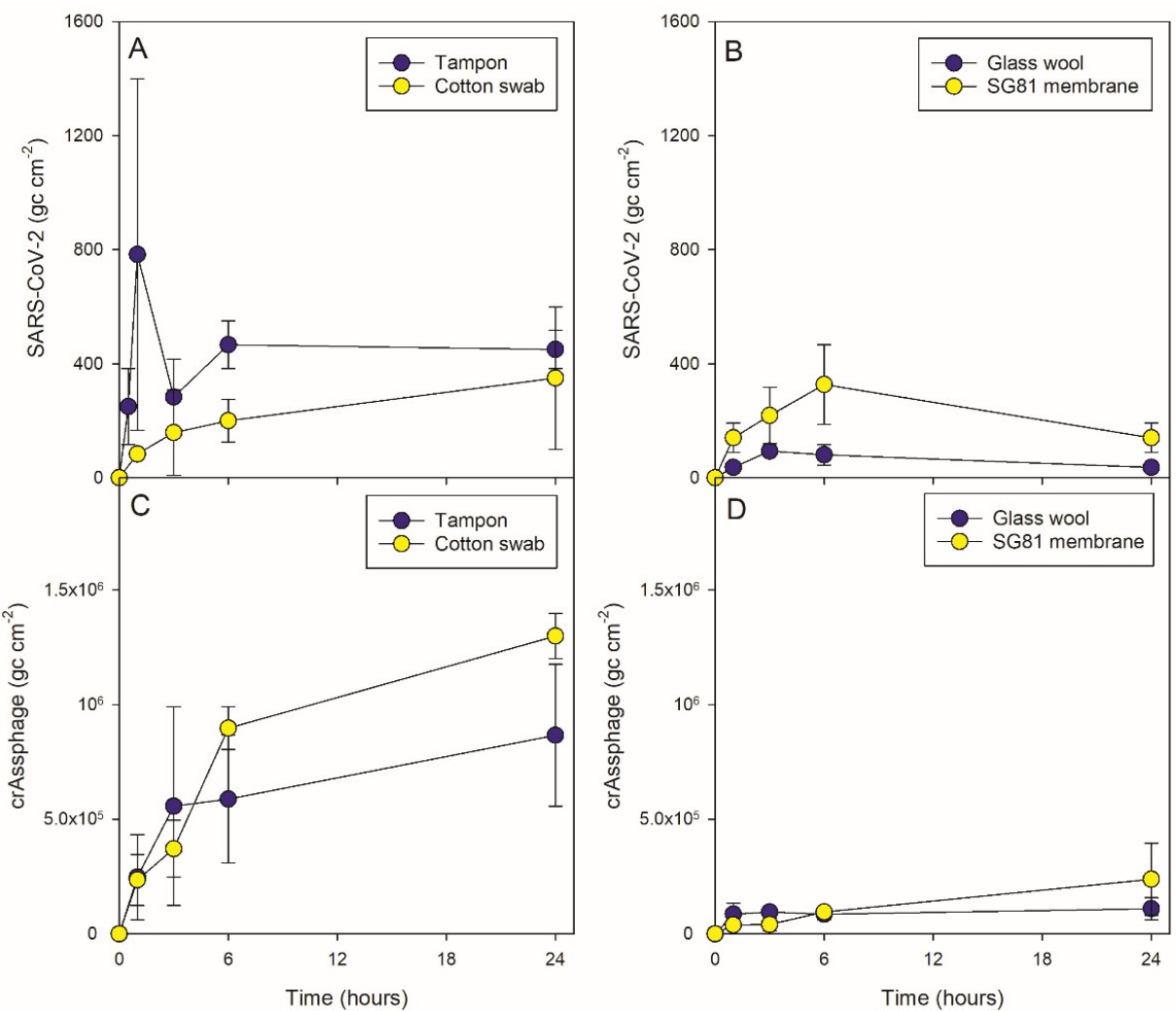

**Figure 4.** Assessment of exposure time (0–24 h) on the accumulation of (**A,B**) SARS-CoV-2 (upper panels) and (**C,D**) the human faecal marker virus crAssphage from four different passive samplers after deployment at a centralized wastewater treatment plant. Values represent means ± SEM (*n* = 3). The two sets of graph panels represent different experimental days and therefore cannot be directly compared.

## 4. Discussion

### 4.1. Charge Properties of SARS-CoV-2 in Relation to Capture by Passive Sampler Materials

The design of the most effective passive sampler requires a good knowledge of the properties that regulate viral sorption to surfaces, particularly in aqueous environments. The ability of SARS-CoV-2 to bind to surfaces is largely dependent on the size, abundance and composition of the outward facing surface proteins (i.e., spike proteins; [56]). Current evidence suggests that the overall diameter of SARS-CoV-2 ranges from 60–140 nm, with spikes that extend ~19 nm from the shell [57]. The exact size of the SARS-CoV-2 virion, however, depends on the cellular conditions in which it was assembled (which affects the protein/lipid ratio; [58]). The size of SARS-CoV-2 suggests that it can enter most pores in commercial filter devices but cannot enter the small pores present in individual cellulose fibers (4–50 nm; [59]).

All SARS-CoV-2 viral particles are charged on their surface and can bind to surfaces through electrostatic, hydrophobic and van der Waals interactions [56,60]. At neutral pH

(e.g., close to wastewater), most viral particles including SARS-CoV-2 are expected to carry a net negative charge as they have an isoelectric point below pH 7 [61,62]. However, multiple patches of positive and negative charge exist in domains of the spike protein [63–66]; Figure S13). However, the overall net charge on the spike protein is expected to be −1.4 at pH 7.4, reflective of wastewater [67]. As the RNA encoding the spike protein continues to mutate and evolve in SARS-CoV-2, these properties are not static and consequently viral lineages may vary in their surface properties allowing them to differentially interact with both charged and hydrophobic surfaces [62,64,68]. Thus, it is likely that intact virus particles can be captured by both electronegative and electropositive passive sampler materials [69–71], but that electropositive materials may be more efficient.

Damaged capsids and free viral RNA released from SARS-CoV-2 particles may also be present in wastewater [72]. In addition, if viral particles are damaged some of the interior proteins may be exposed. From inside to out there is a gradual fall in the isoelectric point from 9.59 in the nucleocapsid to 5.32 in the spike proteins. This is driven by variations in the abundance of $-NH_2$, $-NH_3^+$, $-COOH$, and $-COO^-$ groups in the amino acids that make up these proteins. This implies that at typical wastewater pH values, a virion has a positively charged core surrounded by a negatively charged envelope and spikes [65]. In contrast, studies indicate that free RNA will be highly negatively charged, possessing an isoelectric point of between 5.2 to 6.7 and, therefore, may be captured more favorably by electropositive passive sampler materials [62,73,74]. At present, however, most studies have used electronegatively charged membranes [23,71–75] with no direct comparisons with electropositive membranes undertaken to date to the authors' knowledge.

### 4.2. Success of Different Passive Sampler Materials to Capture SARS-CoV-2 from Wastewater

In this study, a range of different passive sampler materials previously used for viral capture were directly compared. Unsurprisingly, they showed considerable variation in their physicochemical properties (Tables 2–4), which we assume led to their different abilities to capture viruses from wastewater. Despite the pH of the materials ranging between pH 4.1–10.7, we found that the point of zero charge (PZC) for all materials was below pH 7.0, with the exception of the glass wool. This indicates that most of the materials will be negatively charged at pH values typical of wastewater (pH 7.0–8.0; Figure S14). Based on the results of the passive sampler trials performed under both laboratory and field-based conditions, we confirmed that both electropositive and electronegative passive sampler membranes had the potential to capture viruses from wastewater. In a holistic evaluation of each passive sampler performance (Table 5), we conclude that cotton-based products provide a simple and effective way to capture viruses from wastewater. Glass wool and SG81 filters are not recommended. If they are to be deployed, they should only be used in low flow environments to prevent physical degradation or loss of material, as we found these materials disintegrated when deployed in high wastewater flows. In addition, the manual handling of glass wool is also problematic from a human exposure perspective [76,77], as it is easily broken down and can become embedded into the user's skin. We also noted that other passive samplers, such as NanoCeram and ZetaPlus filters were ineffectual or performed poorly relative to their high cost.

The recovery of viruses in laboratory conditions show that there was a significant difference between sampler materials. The cotton-based passive samplers yielded the higher viral recoveries. In our static incubation system (i.e., aerated, but non-flowing), we found that increasing the incubation time from 1 h to 24 h did not increase the retention of Phi6 by any of the materials (data not presented). This suggests that there is either a finite number of sorption sites on the material surface, or more likely that Phi6 is degrading, particularly at the incubation temperature used here [78,79]. Interestingly, analysis of the external bathing solution after 1 h indicated that the ZetaPlus had removed almost all of the Phi6 from solution, however, this could not be extracted from the surface of the material and thus returned a negative result. Further work is therefore needed to establish the correct conditions for successfully desorbing the virus from this surface.

**Table 5.** Summary of the overall performance and ranking of the eight different passive sampler materials when deployed in a wastewater environment.

| Scoring Criteria | Nano Ceram | SG81, Whatman | GW-40 Glass Wool | ZetaPlus, 3M | Cotton-Moore Swab | Biodyne Nylon | 11106, Sartorius | Cotton Tampon |
|---|---|---|---|---|---|---|---|---|
| Code | A | B | C | D | E | F | G | H |
| Viral capture/recovery | | | | | | | | |
| Bacteria capture/recovery | | | | | | | | |
| Ease of use | | | | | | | | |
| Placement in devices | | | | | | | | |
| Multifunctionality | | | | | | | | |
| Particle trapping | | | | | | | | |
| Physical integrity | | | | | | | | |
| Health and safety issues | | | | | | | | |
| Ragging potential | | | | | | | | |
| Cost | | | | | | | | |
| Overall score | 31 | 41 | 39 | 36 | 46 | 38 | 38 | 46 |
| Ranking | 8 | 3 | 4 | 7 | 2 | =5 | =5 | 1 |

**Score**
1 — Low
2
3 — Medium
4
5 — High

In this study, we opted for a simple operational way to recover the virus from the filter (i.e., immersion in lysis buffer), however, other studies have reported that phosphate buffer at high pH values (pH >9.0) can also be effective as an elution buffer [80]. It is possible that ZetaPlus binds enveloped viruses more strongly than their non-enveloped counterparts, however, more work is needed to optimize the recovery of viruses from this material. It should be noted, that previous studies [81,82] also showed that ZetaPlus was poor at recovering bacteriophage from water, although it is not clear if this was due to poor viral sorption or desorption.

The materials with the highest SARS-CoV-2 recovery were the tampon material, glass wool and the SG81 filter. The recovery of SARS-CoV-2 RNA in the 24 h spiked incubations was significantly lower than in the 1 h incubations again suggesting that longer incubation times do not improve recovery in a static incubation system. This poor recovery of SARS-CoV-2 in the bathing media after 24 h also suggests that the virus has degraded, consistent with previous findings [83,84] who also reported a rapid loss of SARS-CoV-2 signal from wastewater (T90 values of ca. 24 h). Conversely, in the field experiments the cotton swabs and tampon passive samplers were ranked the overall best performing for viral capture. However, short deployment times (<6 h) are also recommended to ensure that the passive sampler does not become saturated with particulates present in the sewer. Further work is needed to ascertain if passive samplers can capture other viruses or bacteria of public health interest (e.g., influenza, norovirus, anti-microbial resistance strains). However, the capture of crAssphage and Phi6 provides evidence of the potential for passive samplers to be used to capture other viruses which may be of public health concern. The current literature partially explores novel passive samplers for the capture of viruses (such as enterovirus, human adenovirus (hAdV), and pepper mild mottle virus (PMMoV) in addition to SARS-CoV-2 from wastewater [70], however, further work is needed to explore the effect of passive sampler type upon viral recovery, in addition to viral recovery method from passive sampler (e.g., inclusion of a bead beating step, choice and volume of elution buffer). This is particularly needed for the ZetaPlus membrane where viral capture was good, but subsequent recovery was poor.

The capture pattern of crAssphage was similar to that of SARS-CoV-2, which implies that crAssphage is suitable for normalizing the SARS-CoV-2 data for faecal exposure [85]. Unfortunately, the capture of $NH_4^+$ and P by the passive samplers did not appear to provide a reliable estimate of faecal/urine exposure. Further work should look at the integration of a dedicated ion exchange membrane specifically targeted at capturing N and P for potential normalization. Further work could also investigate chemical modification of the material to promote a greater surface charge density on the passive sampler to promote greater viral capture [86].

In addition to the viral work, we also assessed the suitability of passive samplers for profiling the bacterial community composition. Interestingly we found that NanoCeram passive filter materials gave a very different bacterial profile to that of the other passive sampler materials, whilst the remaining passive samplers produced profiles in line with those often reported in human fecal matter [87,88]. Deployment time had a significant influence on the composition of bacterial (microbial) communities when comparing the 1 and 24 h profiles. As we did not collect a 24 h composite wastewater sample we do not know whether this was due to changes in the microbiome of the wastewater which is expected to change during the day [89], or to the variation in the composition of microbial biofilm communities on the passive sampler surface [90]. In a WBE context, further work could focus on exploring the diversity of antimicrobial resistant strains captured by these different passive sampler materials.

## 5. Conclusions and Future Perspectives

We show that viral binding can occur upon several passive sampler materials, although cotton-based passive sampler materials proved to be consistently the best for viral capture under both laboratory and field experiments with an optimal 6 h exposure time under

moderate to high flow conditions. Furthermore, we demonstrate that passive samplers can be simultaneously used for investigating bacterial community structure. However, whilst undertaking this work a series of areas for further research were identified as follows:

1.  In this study we use qPCR-based approaches for the quantification of SARS-CoV-2 and the faecal marker virus, crAssphage. While successful, it would be useful to interface the passive samplers with other analysis approaches (e.g., field-based RT-LAMP), to assess their suitability for remote deployment.
2.  There is increasing interest in the analysis of SARS-CoV-2 variants of concern or interest in wastewater. Additional research is therefore required to investigate the quality of RNA recovered from passive samplers in comparison to that recovered from refrigerated autosamplers.
3.  The trials detailed here were undertaken in well mixed wastewater at central wastewater treatment works. Further trials are required using a range of materials at close-to-source locations (e.g., prisons, residential blocks, hospitals) where the wastewater is more temporally and spatially (more intact faecal material) heterogenous.
4.  Further tests are needed to ascertain the relative contribution of particulate trapping on the retention of SARS-CoV-2 by passive samplers relative to that electrostatically held to the passive sampler surface. In this context, a direct comparison is needed between electropositive and electronegative sampler materials at retaining viruses and their associated genetic material.

**Supplementary Materials:** The following supporting information can be downloaded at: https://www.mdpi.com/article/10.3390/w14213568/s1, Figure S1 to Figure S14 and Table S1 to Table S2. Reference [91] is cited within Supplementary material.

**Author Contributions:** R.W., M.J.W., J.M.S.G., D.L.J., A.C.S. and K.F. conceived the project. D.L.J., C.P., K.L.-S., G.B.W., R.W.B. and K.F. undertook the experimental work. D.L.J., C.P., R.W.B. and K.F. undertook the processing and analysis of the data. R.B. undertook the processing and analysis of the sequencing data. D.L.J. led the data interpretation. D.L.J. and J.L.K. led the writing of the manuscript. All authors have read and agreed to the published version of the manuscript.

**Funding:** This research was funded by the UK Joint Biosecurity Centre and the Department of Health and Social Care under the ACE Next Gen Wastewater Based Epidemiology C215.2 programme. The Centre for Environmental Biotechnology Project is funded though the European Regional Development Fund (ERDF) by Welsh Government.

**Data Availability Statement:** Data is available on request from the authors.

**Acknowledgments:** We particularly thank Tony Harrington at Dŵr Cymru Welsh Water alongside staff at the wastewater treatment facilities for their support in this project. We thank Constance Tulloch, Sarah Chesworth, Emily Cooledge and Daniel Robinson for technical support.

**Conflicts of Interest:** The authors declare no conflict of interest. The funders had no role in the design of the study; in the collection, analyses, or interpretation of data; in the writing of the manuscript; or in the decision to publish the results.

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
