# Peer review of "Critical Evaluation of Different Passive Sampler Materials and Approaches for the Recovery of SARS-CoV-2, Faecal-Indicator Viruses and Bacteria from Wastewater"

_water, doi:10.3390/w14213568_

Round 1

Reviewer 1 Report

MS_Nº: water-1979600

Title: Critical Evaluation of Different Passive Sampler Materials and Approaches for the Recovery of SARS-CoV-2, Faecal-Indicator Viruses and Bacteria from Wastewater

Authors: Davey L. Jones, Jasmine M.S. Grimsley, Jessica L. Kevill, Rachel Williams, Cameron Pellett, Kathryn Lambert-Slosarska, Andrew C. Singer, Gwion B. Williams, Rafael Bargiela, Robert W. Brown and Kata Farkas

START of REVIEW

The MS by Jones and coworkers is a solid piece of work that provides a comprehensive evaluation of different materials routinely used as passive samplers for wastewater-based epidemiological studies. The authors assessed the performance of the different materials selected for the capturing of SARS-CoV-2 and Phi6 viruses as well as their stability once deployed into wastewater. Indeed, the authors carried out exhaustive work covering different aspects such as material composition, physical and chemical characteristics of the collected materials, physical integrity, amount of particulate matter collected, recovery of target viruses, and the capacity of the studied materials to provide a valid profile of the bacterial community composition. Besides, the work is timely considering the current implementation of surveillance programs to track the communal circulation of SARS-CoV-2 and the attention that this approach has gained in the last two years to track the circulation of SARS-CoV-2 at different spatial scales. To be honest, I really enjoyed reading the MS. Thus, my recommendation is to accept the MS for publication. However, some minor issues should be carefully addressed before final acceptance.

SPECIFIC COMMENTS

- Abstract, L20: “passive samplers”, plural?

- Abstract, L25: “…the bacteriophage Phi6”

- Abstract, L33: 16S profile is jargon. Please, use “the 16S rRNA gene profile” or, alternatively, refer to the bacterial composition.

- L41: The first sentence should be deleted since it seems a piece of advice to authors on what the introduction section is about.

- L44: “attraction”, I presume.

- L46: In general, I’m against the use of “levels” as synonyms for “concentrations”. I thus suggest using the latter term here.

- L91: physicochemical is misspelled.

- L176: Figure S2 shows the location of three WWTP but the authors only show results from one of them. Although this is stated in the last sentence of the figure caption, I suggest redrawing the figure to show only the location of the studied WWTP. In the current design, it seems that the authors are reusing a figure from another paper.

– L177: At all sites? Only one WWTP, right?

– L281: the title of this subsection is confusing since 16S is linked to the section numbers. I suggest replacing the current title with “DNA extraction, 16S rRNA gene amplicon sequencing and analysis”

– L367: Same comment here, the 16S is linked to subsection numbers. Here I suggest replacing the current title with “Profiling bacterial community composition from different…”

– L368: 16S rRNA gene profile

– L375: It is not clear to me what Halobacteria the authors refer to. In fact, this name is outdated since Halobacteria are now classified as Haloarchaea. On the other hand, Halobacteria doesn’t appear in Figure 3 since the only archaeal taxon is Nanoarchaeota, which is a phylum within the superphylum DPANN. Please, correct.

– L379: Regarding Figure 3: I suggest replotting the figure using other colors since the current legend contains only 7 colors for 34 taxa, causing a repetition of colors for some taxa (e.g., green is used for Gammas, Actinos, Leptos, Synergistia, and an environmental metagenome. Also, I suggest using a colorblind-friendly color palette.

– L446–449: Again, these sentences refer to indications for authors. Please, delete.

– L545: antimicrobial-resistant strains

– L563: Here, I suggest: “…the suitability of passive samplers for profiling the bacterial community composition.”

– L568: 16S is jargon. Here, however, I would suggest: “…significant influence on the composition of bacterial (microbial) communities when comparing…”

– L571: The use of “progressive evolution” could mislead the reader. I suggest replacing it with “…or to the variation in the composition of microbial biofilm communities…”

– L580: 16S rRNA gene profiling

– L576–599: In my opinion, this final section is more a perspective on future studies and research needs than a summary of the main conclusions of the work. I suggest compiling first the main conclusions of the work and then identifying the main research needs. Alternatively, just change the title of the section.

– L600: This table deserves a better design. First, use a colorblind-friendly color palette. Second, I suggest using the same letter codes for tested materials as those used, for instance, in figures 2 and 3.

Reviewer 2 Report

This paper compared different passive sampler materials for the recovery of viruses (SARS-COV-2, faecal-indicator) and bacteria from wastewater as cheap tools to be used for pathogen detection in wastewater.

The paper is well written and the methods are well conducted.

However, authors should explain more the originality of this work since  most of the tested passive sampler materials were previously investigated for their ability to recover viruses (SARS-COV-2) and bacteria from wastewater.

In addition, the aim of pathogen monitoring is to early detect infections which requires sampling methods and materials with high sensitivity  to detect pathogens in wastewater from low case number settings. In this study, SARS-COV-2 viruses  were not detected during the period of the experiment characterized with  low rate of COVID-19 cases. Have authors tested the  sensitivity of these sampler materials to determine the  lowest detectable  SARS-COV-2 gc L−1  in wastewater?

Please remove the first sentence in the introduction and add few sentences to demonstrate the originality of this study as compared to the previous works of the authors and the literature.
